

# Ca²⁺ dynamics in zebrafish morphogenesis

Yusuke Tsuruwaka[1,*], Eriko Shimada[1,2,3,*], Kenta Tsutsui[1] and Tomohisa Ogawa[1]

[1] Marine Bioresource Exploration Research Group, Japan Agency for Marine-Earth Science and Technology (JAMSTEC), Yokosuka, Japan
[2] Department of Animal Science, University of California, Davis, CA, United States
[3] Cellevolt, Yokohama, Japan
[*] These authors contributed equally to this work.

## ABSTRACT

Intracellular calcium ion (Ca²⁺) signaling is heavily involved in development, as illustrated by the use of a number of Ca²⁺ indicators. However, continuous Ca²⁺ patterns during morphogenesis have not yet been studied using fluorescence resonance energy transfer to track the Ca²⁺ sensor. In the present study, we monitored Ca²⁺ levels during zebrafish morphogenesis and differentiation with yellow cameleon, YC2.12. Our results show not only clear changes in Ca²⁺ levels but also continuous Ca²⁺ patterns at 24 hpf and later periods for the first time. Serial Ca²⁺dynamics during early pharyngula period (Prim-5-20; 24–33 hpf) was successfully observed with cameleon, which have not reported anywhere yet. In fact, high Ca²⁺ level occurred concurrently with hindbrain development in segmentation and pharyngula periods. Ca²⁺ patterns in the late gastrula through segmentation periods which were obtained with cameleon, were similar to those obtained previously with other Ca²⁺sensor. Our results suggested that the use of various Ca²⁺ sensors may lead to novel findings in studies of Ca²⁺ dynamics. We hope that these results will prove valuable for further research in Ca²⁺ signaling.

## INTRODUCTION

Intracellular calcium ions (Ca²⁺) act as second messengers in organism cellular signaling pathways. Ca²⁺ is relevant to most biological phenomena, and is particularly relevant to early development (*Niki et al., 1996*; *Berridge, Lipp & Bootman, 2000*; *Slusarski & Pelegri, 2007*). Patterning intracellular Ca²⁺ concentration is important for the study of living organisms. Ca²⁺ has been measured using aequorin since the late 1960s, and using fluorescent proteins such as modified green fluorescent protein since the late 1990s (*Shimomura, Johnson & Saiga, 1963*; *Miyawaki et al., 1999*; *Takahashi et al., 1999*). To date, Ca²⁺ patterns during zebrafish development have been studied mostly using aequorin, and many patterns have been described (*Créton, Speksnijder & Jaffe, 1998*; *Jaffe, 1999*; *Webb, Chan & Miller, 2013*). However, to image Ca²⁺ patterns in more detail, a multifaceted analysis with a variety of chemical indicators is required. Advantage of a luminescent Ca²⁺ sensor such as aequorin is that not carrying phototoxicity due to excitation lights. On the other hand, disadvantages are (1) requirement of the substrate coelenterazine which is gradually consumed, (2) difficulty of detecting subtle signals which is weaker than the one fluorescent

Corresponding author
Yusuke Tsuruwaka,
tsuruwaka@jamstec.go.jp

$Ca^{2+}$ sensor emits, (3) occasionally unsuitable for a long-term and high-speed photography. To present, 'continuous' $Ca^{2+}$ patterns such as long-term time lapse imaging in zebrafish morphogenesis after 24 hpf (hour post fertilization) have not been reported yet. Meanwhile, stable $Ca^{2+}$ signals are expected with fluorescent $Ca^{2+}$ sensors such as yellow cameleon YC2.12 because the sensor molecule is integrated into cells. This is advantageous in long-term measuring since $Ca^{2+}$ sensor is synthesized *in vivo* and does not require a substrate like luminescent $Ca^{2+}$ sensor does. Fluorescence emits stronger light than luminescence in general although requiring an excitation light, which enables us to measure real-time and to detect subtle signals.

Recently, we also reported that morphological changes which had been the consequences of *wwox* gene down regulation by morpholino injection brought about dramatic transition in $Ca^{2+}$ signaling (*Tsuruwaka, Konishi & Shimada, 2015*). To date, with cameleon consecutive $Ca^{2+}$ dynamics of zebrafish gastrulation was reported (*Tsuruwaka et al., 2007*). The purpose of the present study was to analyze serial $Ca^{2+}$ patterns for long-term periods, from late gastrula to pharyngula periods, using cameleon.

## MATERIALS AND METHODS

### Zebrafish and $Ca^{2+}$ imaging

Experiments were conducted as previously described (*Tsuruwaka et al., 2007*; *Tsuruwaka, Konishi & Shimada, 2015*). Briefly, 3 nL of synthetic YC 2.12 mRNA (0.5 ng/mL) was injected into blastodiscs of each single-cell embryo. After YC2.12 had confirmed to be distributed ubiquitously in the whole embryo, FRET analyses were performed as followed. Fluorescence images were obtained using a Zeiss Axiovert 200 microscope equipped with a combination of two filters, i.e., CFP-CFP, YFP-YFP, and CFP-YFP filters (Carl Zeiss, Oberkochen, Germany). Amplification and numerical aperture of the objective lens were 5× and 0.16, respectively. An AxioCam MRc5 camera (Carl Zeiss) was used to photograph the images, and the image analysis was performed using Axiovert FRET version 4.4 software (Carl Zeiss). Fluorescence was quantified following the manufacturer's instructions. The control experiment was performed using $Ca^{2+}$-ATPase inhibitor thapsigargin (Wako Pure Chemical Industries, Osaka, Japan) to confirm YC2.12 would work correctly (*Schneider et al., 2008*; *Popgeorgiev et al., 2011*). The number of eggs analyzed was 300 each experiment and the experiments were performed for total 37 times. Of those, 50 eggs were employed in the control experiment. No approval was required to conduct studies on fish according to the Ministry of Education, Culture, Sports, Science and Technology, Notice No. 71 (in effect since June 1, 2006).

## RESULTS AND DISCUSSION

### $Ca^{2+}$ dynamics during zebrafish morphogenesis

$Ca^{2+}$ patterns showed dynamic changes during zebrafish morphogenesis (Fig. 1). Since the $Ca^{2+}$ monitoring had been well studied with aquorin by *Créton, Speksnijder & Jaffe (1998)*, we mainly focused on novel findings here. High $Ca^{2+}$ levels were observed in the anterior and posterior body regions from stages bud to 16-somite (10–17 hpf). In the anterior

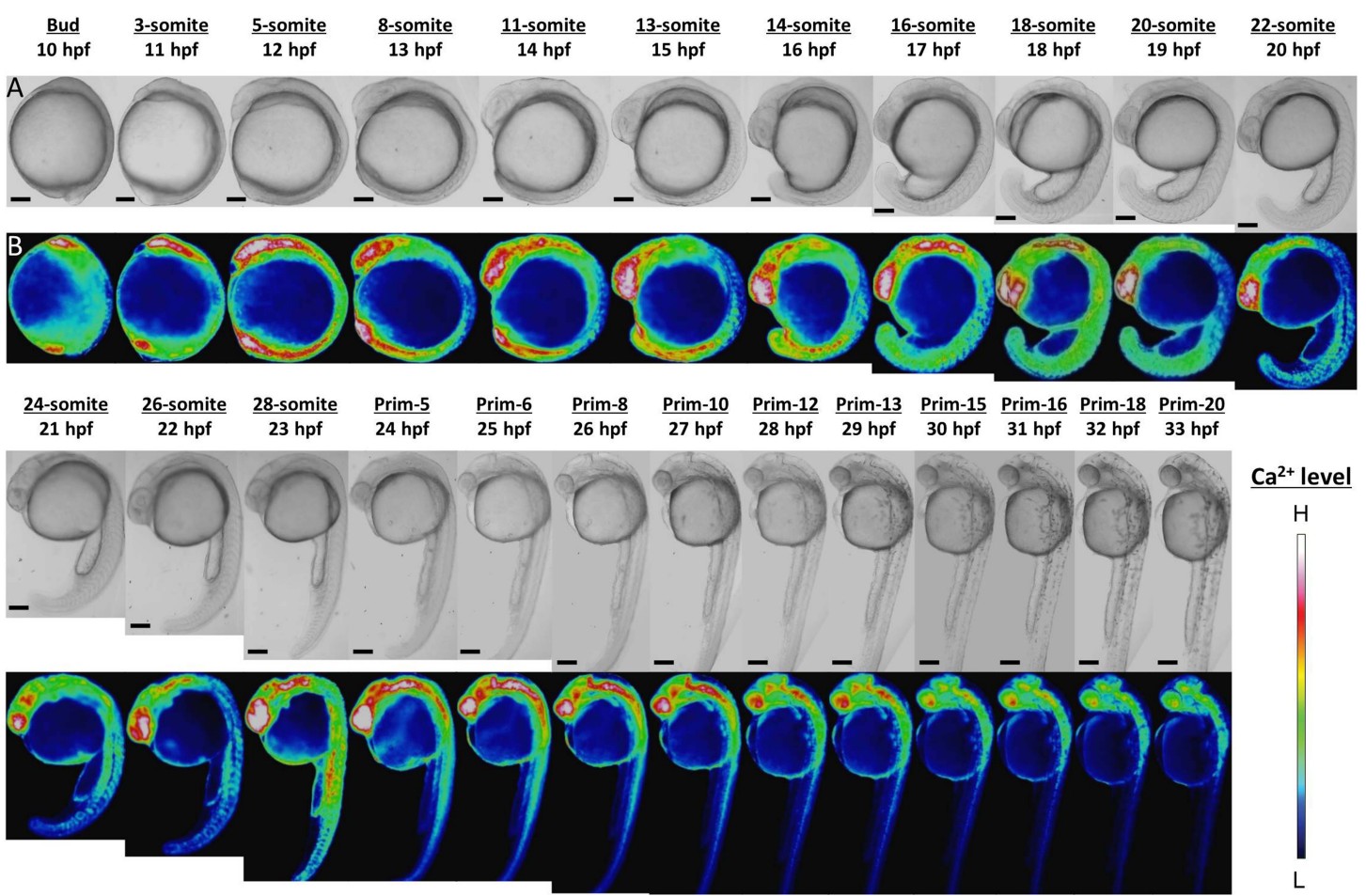

**Figure 1 Ca²⁺ dynamics in the late gastrula, segmentation, and early pharyngula periods.** (A) Bright field image; (B) color-coded image; scale bar, 200 μm (magnification, ×50). The color-coded image shows Ca²⁺ levels as white (high Ca²⁺) and blue (low Ca²⁺). Embryos used in this experiment demonstrated normal development and grew to adulthood.

trunk, the Ca²⁺ level reached a peak at 18-somite stage, whereas in the posterior trunk the Ca²⁺ peak was shown at 28-somite stage (Fig. S1).

In the developing head, the high level of Ca²⁺ was maintained through to prim-13 stage. Notably, this high Ca²⁺ level occurred concurrently with development of rhombomere, a segment of the developing hindbrain, from stages 26-somite to prim-10 (Fig. S2). Ca²⁺ level at presumptive midbrain increased at 26-somite stage and reached maximum level at prim-5 stage. Moreover, Ca²⁺ concentration at presumptive rhombomere 2 and 4 in hindbrain started to rise from 26-somite stage and then all rhombomeres showed relatively high Ca²⁺ levels at prim-5 stage. Ca²⁺ at rhombomere 2 reached maximum level at prim-5 stage, whereas rhombomere 1, 3 and 4 did at prim-6. With focusing on the rhombomere and midbrain hindbrain boundary (MHB), it is quite interesting to consider relevance between Ca²⁺ signals and formation of neuronal network. Ca²⁺ involves with neural network in zebrafish and Ca²⁺ sensors were used for studying neuronal activity and reflexive behavior (*Higashijima et al., 2003*; *Muto et al., 2013*; *Portugues et al., 2014*). Serial

neural circuits such as sensory neuron, intercalated neuron, motor neuron, muscle were formed within 24 hpf in zebrafish (*Saint-Amant & Drapeau, 1998*; *Downes & Granato, 2006*; *Fetcho, Higashijima & McLean, 2008*; *Pietri et al., 2009*). When those circuits become active, zebrafish acquires stimulus-response. High $Ca^{2+}$ levels at trunk and rhombomere regions in our results coincide with the development and activation of those circuits. Especially, Mauthner cells at rhombomere 4 become active and stimulate neural circuits, which results in triggering various body movements such as escape behavior (*Korn & Faber, 2005*). In fact, rhombomere and MHB during brain organization closely involved with Wnt signaling pathway which controls $Ca^{2+}$ signaling (*Webb & Miller, 2000*; *Prakash & Wurst, 2006*). Therefore, $Ca^{2+}$ dynamics at developing head in our results suggested intimate correlation with and formation and activation of neural circuits.

In the developing tail, the $Ca^{2+}$ level had dropped by 20-somite stage and stabilized at a low level. The patterns in $Ca^{2+}$ levels through the late gastrula and segmentation periods (Bud-28-somite stages; 10–23 hpf) that we obtained with yellow cameleon, YC2.12, were similar to those obtained previously with aequorin (*Créton, Speksnijder & Jaffe, 1998*; *Webb & Miller, 2000*). However, we succeeded in observing $Ca^{2+}$ patterns during early pharyngula period (Prim-5-20; 24–33 hpf) which have not reported anywhere yet.

Correlations between zebrafish morphogenesis and intracellular $Ca^{2+}$ dynamics in the late gastrula-segmentation periods have been well characterized by Webb, Miller and colleagues (*Gilland et al., 1999*; *Webb & Miller, 2007*). Their work on $Ca^{2+}$ dynamics during somitogenesis is particularly informative (*Webb & Miller, 2010*; *Cheung et al., 2011*; *Webb et al., 2012*).

Our finding of increasing $Ca^{2+}$ levels in the anterior region during the pharyngula period, when the basic body plan is complete, is consistent with $Ca^{2+}$-related gene expression, which controls the formation of the brain and nervous system (*Zhou et al., 2008*; *Hsu & Tseng, 2010*). Moreover, patterns of CaMK-II gene expression are in agreement with our observations of $Ca^{2+}$ patterns at 3-somite, 18-somite, prim-5 stages and later, suggesting that this gene is closely involved with $Ca^{2+}$ dynamics (*Rothschild, Lister & Tombes, 2007*). Figure S3 showed the compared images between our results and the CaMK-II expressions based on *Rothschild, Lister & Tombes (2007)*. In fact, *Freisinger et al. (2008)* discuss correlations between $Ca^{2+}$ signaling pathways and zebrafish body plan formation. The present study showed that cameleon, a genetically encoded $Ca^{2+}$ sensor, enables us to analyze $Ca^{2+}$ dynamics clearly during development and differentiation in a zebrafish embryo. YC2.12 worked correctly as $Ca^{2+}$ sensor in whole living embryos since treatment with $Ca^{2+}$-ATPase inhibitor thapsigargin induced altered $Ca^{2+}$ level (Fig. S4). The embryo shown in Fig. S4B exhibited the increased $Ca^{2+}$ level at later stages, which was consistent with the results reported by *Popgeorgiev et al. (2011)* (data not shown). We have achieved in tracking the serial $Ca^{2+}$ patterns from late gastrula to early pharyngula periods for the first time. This use of a variety of $Ca^{2+}$ sensors has led to a novel perspective in the study of $Ca^{2+}$ dynamics.

In future, tracking whole body $Ca^{2+}$ signaling patterns with cameleon in addition to aequorin and other sensors may provide even more detail on $Ca^{2+}$ signaling during zebrafish development. Thus, instead of discussing whether some $Ca^{2+}$ indicators are superior to others, we propose that the use of a variety of indicators may give better results.

Further comparison of our cameleon study results with those of previous $Ca^{2+}$ studies should lead to more insight into $Ca^{2+}$ dynamics.

## CONCLUSIONS

$Ca^{2+}$ patterns showed dynamic changes during zebrafish morphogenesis, as illustrated using cameleon, a genetically encoded $Ca^{2+}$ sensor. Continuous $Ca^{2+}$ dynamics observed with cameleon at 24 hpf and later periods was investigated for the first time. The results suggested that the use of a variety of $Ca^{2+}$ sensors may lead to novel findings in studies of $Ca^{2+}$ dynamics.

## ACKNOWLEDGEMENTS

We would like to thank Dr. Atsushi Miyawaki for providing the YC2.12 construct and Dr. Takafumi Konishi for his helpful advice. We also would like to thank anonymous reviewer for the helpful suggestions.

### Funding

The authors received no funding for this work.

### Competing Interests

The authors declare there are no competing interests.

### Author Contributions

- Yusuke Tsuruwaka conceived and designed the experiments, performed the experiments, analyzed the data, contributed reagents/materials/analysis tools, wrote the paper, prepared figures and/or tables, reviewed drafts of the paper.
- Eriko Shimada performed the experiments, analyzed the data, wrote the paper, prepared figures and/or tables, reviewed drafts of the paper.
- Kenta Tsutsui and Tomohisa Ogawa performed the experiments, analyzed the data.

### Animal Ethics

The following information was supplied relating to ethical approvals (i.e., approving body and any reference numbers):

No approval was required to conduct studies on fish according to the Ministry of Education, Culture, Sports, Science and Technology, Notice No. 71 (in effect since June 1, 2006).

### Data Availability

The raw data has been supplied as Data S1.

### Supplemental Information

Supplemental information for this article can be found online at http://dx.doi.org/10.7717/peerj.2894#supplemental-information.

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
