# Peer review of "Ca2+ dynamics in zebrafish morphogenesis"

_PeerJ, doi:10.7717/peerj.2894_

## Round 0.1 · original submission · Major Revisions

· Academic Editor

Major Revisions

As you can see from the reviewers' comments, the work as it stands now is wanting in several important respects. I would like to point out that, if the authors choose to address the reviewers' concerns, especially reviewer #2, the manuscript would need a significant reworking of images, data, and clarification of several points, such as reproducibility and novelty with respect to other methods and similar results already available.

·

Basic reporting

The article is well written. The introduction is sufficient and properly highlights advantages and disadvantages of the different intracellular calcium sensors. The figure is relevant for the content of the article and is properly labeled. However a phrase in lines 82 to 84 is confusing since it states “In the dorsal area, the Ca2+ level reached a peak between the 14- and 18-somite stages”. It is not clear what they mean with dorsal area. The intense signal is clearly form the eyes and what appear to be the mesencephalon and most anterior half of the hindbrain. More detailed analysis is required in this point.

Experimental design

No Comments

Validity of the findings

1) There is no information on how many repetitions of the experiment were performed, since this is an indication of the observed patterns reproducibility.
2) A relevant detail is missing. The objective characteristics like amplification and numerical aperture.
3) The approach is interesting. However I consider that some control experiments should be included. For example there are several Ca2+ pharmacological modulators that have been used in zebrafish [1] compounds that induce calcium release in vivo to address the specificity of the sensor in zebrafish at the analyzed stages. Unfortunately these controls were not described in the previous reports from the same group [2, 3] that submitted the present manuscript. Part of the controls could have been done when this particular sensor was originally developed, however most of these controls are performed in cell lines like HeLa but when these are expressed in whole embryos the results should be taken with caution since artifacts can arise in whole living organisms.
4) In this line of thinking I couldn’t find the main reason why they choose to use calcium sensor YC2.12, what characteristics were taken into account compared with other YC sensors (calcium sensitivity, dynamic range, signal-to noise ratio, etc.) PeerJ has a broad audience and this information will improve the understanding and potentially the impact of this article. The pros and cons compared to aequorin are explained in detail in the main text in the introduction, but this particular information for the YC2.12 sensor can be included.

1. Lam, P.Y., et al., Inhibition of stored Ca2+ release disrupts convergence-related cell movements in the lateral intermediate mesoderm resulting in abnormal positioning and morphology of the pronephric anlagen in intact zebrafish embryos. Dev Growth Differ, 2009. 51(4): p. 429-42.
2. Tsuruwaka, Y., et al., Real-time monitoring of dynamic intracellular Ca(2+) movement during early embryogenesis through expression of yellow cameleon. Zebrafish, 2007. 4(4): p. 253-60.
3. Tsuruwaka, Y., M. Konishi, and E. Shimada, Loss of wwox expression in zebrafish embryos causes edema and alters Ca(2+) dynamics. PeerJ, 2015. 3: p. e727.
4. Nagai, T., et al., Expanded dynamic range of fluorescent indicators for Ca(2+) by circularly permuted yellow fluorescent proteins. Proc Natl Acad Sci U S A, 2004. 101(29): p. 10554-9.
5. Mizuno, H., et al., Transgenic zebrafish for ratiometric imaging of cytosolic and mitochondrial Ca2+ response in teleost embryo. Cell Calcium, 2013. 54(3): p. 236-45.

Comments for the author

1) I wonder why the authors did not use confocal microscopy (spinning disk), since the image quality obtained with this type of microscopy in general have much better spatial resolution. Confocal microscopy was used by the group that originally developed the YC2.12 and other Ca2+ sensors [4] and even in a latter report that included the visualization in 24-hour embryos [5].
2) Images in some regions are to some degree saturated and this lead to spatial and temporal information loss.
3) Perhaps higher magnification images from regions of particular interest, like the brain, will provide more detail on the sensor patterns.

References are in the previous section.

Reviewer 2 ·

Basic reporting

This is a short manuscript that reports calcium dynamics from the Bud stage to 33 hpf of development in zebrafish, the authors use as a reporter the yellow cameleon calcium sensor protein, injected as an mRNA. In Figure 1 they show calcium higher intensity signal as elevated levels of calcium and the main finding is that this pattern changes as development progress. This calcium sensor has already been reported by this group. Although their data is potentially interesting, I believe this is an incomplete work and I believe further work is needed before is ready for publication in PeerJ. Mainly I see no reason to show interesting data, which has not been deeply analyzed and properly presented in the manuscript.

Results are scarcely described and there is more important information in the embryos showed in the figure than the authors mention and it will be an important contribution to the field if these embryos and their calcium signal were analyzed in more detail, for example using close ups at higher amplification and showing some time lapse movies.

In this text there is a general lack for detail in their descriptions for their own results. For example; while the authors point out that calcium elevated levels may coincide with rhombomere formation (28 somites – prim-10), they fail to mention early calcium variations in posterior regions that could be related to somite formation. These has been observed by other groups (Their own citations: Webb & Miller, 2010; Cheung et al., 2011; Webb et al., 2012). As a suggestion they could use arrows in Figure 1 to point to the rhombomere observations and offer some explanation about what this means. The calcium signals during rhombomeres and somite formation should be studied in more detail using higher amplification and time lapse movies. It this is not possible, at least they could improve their figure, showing and discussing these important findings with more detail.

One central problem is that the whole experiment is based on the assumption that YC mRNA, after injection, is expressed in the whole embryo and it is, at some extent, evenly distributed, however this was not proved for this work. Could the authors test that, by ISH or immunostainings of the cameleon sensor protein.?

Specific comments:

The Abstract section is not clear and must be re-written to better describe the work.

The Introduction section is incomplete and inaccurate in some points. It could be important to argue in this section why it is important to carry out studies, using its yellow cameleon sensor, specifically in developmental stages beyond gastrulation. There is not mention of important information about calcium sensor used in neurodevelopment and behavior in zebrafish like (Muto et al 2013, Curr Biol 18:307-11) (Portugues et al 2014 Neuron 19:1328-43). It seems important since the authors wish to expand their observations beyond gastrulation developmental stages. It is not clear how the work fits into the broader field of knowledge.


Lines 36 and 37: This is inaccurate, GFP is not a calcium sensor by itself but modified versions of GFP are.

In the manuscript the sentence “Ca2+ patterns” is repeatedly used, however is not clear for this reviewer what is the meaning of the sentence: The amount of calcium in each cell with respect to other cells during development. The intracellular flux of calcium. Changes in calcium concentrations during different developmental stages.? Please clarify this.

Line 51: …luminescence in genera although… (in general)

Line 55: There is no need to mention that the previous report was published in the same journal, since this information is at the reference itself.

Lines 56 and 57: It is interesting to determine how calcium concentrations change during development and since these embryos are amenable to live imaging techniques could be a tool to study certain cellular dynamics related to the calcium chemistry. However, the authors fail to justify their study and they only mention it was done since there were...a number of requests for more information…

It is not clear in the Materials and Methods section; how many embryos were used in figure 1. Each image is from a different embryo? It is difficult to imagine how they could be so precise, imaging different embryos, in specific developmental stages. For example; imaging prim-15 and prim16, would be much easier to do by imaging the same embryo in a temperature controlled chamber, than using different embryos.
How many times, every image was repeated? It was always consistent?

Results section:

Line 77-79: In posterior regions there seems to be some variations in calcium distribution between Bud and 14 somites stage, however the authors do not report on these. For example, at the 3 somite stage high calcium is only at the bud, while at 13 and 14 somites seems to be a somite specific calcium distribution. Specially since previous reports showed calcium dynamics during somitogenesis (line 91)

The authors should consider to show with more detail (more amplification) calcium detection in somites 12 – 14 somites? More amplification or time lapse movie.

Lines 79-81: Between 26 somites and Prim-13 stages, there are as well fine variations not described in the manuscript. It could be seen that at 26-28 somites there are elevated amounts of calcium at the hindbrain and at some portion of the MHB. It is the MHB, as a brain organizer, regulating calcium levels? Why?

Lines 95-98: The authors describe where is CaMK-II being expressed at the 3-somite, 18-somite and prim-5 stages (according to Rothschild et el). This is one important observations that the authors should point in Figure 1 or in a new diagram, where precisely in the embryo CaMK-II expression patterns and calcium detection match. This will help to follow their Results and Discussion section.

In the Conclusions section:

Lines 113-115: It is a contradictory statement that cameleon and aequorin show similar results and then mention that the use of variety of calcium sensors may provide novel findings.


Figure 1. It should be important to add scale bars in the figure.
Figure 1. Is not appropriately described and labeled.

Experimental design

The experimental design is appropriate, except there is no prove in this work, that mRNA for the Calcium sensor is evenly distributed in the whole embryo.

Validity of the findings

No Comments

Comments for the author

While the findings are interesting they are not presented with enough detail, the manuscript was not prepared with accuracy and have many flaws.

---

## Round 0.2 · Minor Revisions

· Academic Editor

Minor Revisions

Please answer the remaining concerns expressed by reviewer #1.

·

Basic reporting

I think that the authors made a good effort to address most of the concerns indicated in my comments to the manuscript.

Experimental design

No Comments

Validity of the findings

I believe that the specificity of the calcium sensor in whole embryo samples has still not been fully characterized. The embryo presented in the response letter treated with bisphenol A, show changes in camaleon signal patterns. However an image of a non-treated embryo, with an equivalent orientation, should be shown for comparison purposes. The description of the experiment and how the signal patterns change can be included as supplementary information. In my experience to generate intensity decrease with different signaling sensors in whole living embryos, in most cases is difficult and is common to observe sensor pattern changes, as described by the authors. For this reason I suggested to include images of embryos treated with compounds that are known to induce calcium release in vivo. The developmental effects of calcium pharmacological modulators are an expected consequence, due to the calcium signaling importance, but the intensity increase in the sensor signal is likely to occur.

Comments for the author

I think that parts of the added text are confusing.
For example in the introduction the paragraph in lines 61 to 65 imply that you are referring to Ca2+ signaling changes during normal development. However the phrase (lines 61 and 62) "we also reported that morphological changes brought about dramatic transition in Ca2+ signaling" as reported in Tsuruwaka, Konishi & Shimada, 2015, the morphological changes are the consequences of wwox gene down regulation by morpholino injection. Please indicate this in the text.

In the results section
I suggest that in lines 95 to 96, to keep the text as in the previous version “Notably, this high Ca2+ level occurred concurrently with rhombomere development development of rhombomere, a segment of the developing hindbrain, from stages 26-somite” suppressing the text as shown.

The phrase in lines 103 to 104 is not clear what they meant "Ca2+ involves with neural network in zebrafish and Ca2+ sensors were used for studying neuronal activity and reflexive behavior" it should be better explained.

In lines 113 to 114 “Ca2+ movements at developing head in our” instead of movement I suggest to change it for “dynamics”, since you did not show any evidence of calcium movement .

Reviewer 2 ·

Basic reporting

The second version of the manuscripts was greatly improved and I believe it could be now accepted for publication in its current form.

Experimental design

Experimental design is explained more clearly in the second version

Validity of the findings

The findings and the discussion are presented in more clear detail in the second version

Comments for the author

The work, in general, has been improved.

---

## Round 0.3 · Minor Revisions

· Academic Editor

Minor Revisions

Please heed the comments of reviewer #1.

·

Basic reporting

The manuscript shows important improvement

Experimental design

No comments

Validity of the findings

The manuscript shows important improvement. However the control experiments that I asked for since my first evaluation are still not presented. In my opinion the authors using any kind of sensor or performing any type of experimental treatment most characterize their specificity in their own experimental settings and model organism. Simple or direct extrapolation of results and control experiments performed in cell lines or alternative animal models is not adequate. A developing whole embryo is very complex environment that is constantly changing due to tissue remodeling and to the diversity of differentiating cell types. The validity of the results depends on performing accurately planned control experiments.

In the original characterization of cameleon YC2.12 the investigators that developed the sensor, performed controls in cell lines cultured in vitro and in cells as close as possible to a “natural” context in living tissues. The authors reported the treatment of mouse cerebelar slices with a high potassium buffer, an indirect way to affect Ca2+ release. The treatment caused an important FRET signal in Purkinge cells dendritic arbors, which were assumed to indicate a rise in Ca2+(Nagai et al., 2002). Later the same group discussed that despite YC2.12 have a good signal to noise ratio, the sensing domains of YCs may interact with endogenous CaM or CaM-binding proteins decreasing their dynamic range. Based on this potential disadvantage they developed YC2.60 as an alternative (Nagai et al., 2004), which was used in early zebrafish embryos to characterize its dynamics (Mizuno et al., 2013). The observed patterns using YC2.60 or YC2.12 sensors in whole zebrafish embryos show some degree of equivalence to aequorin or Calcium Green dextran sensors (Chang and Lu, 2000; Ma et al., 2009). For these “conventional” sensor molecules control experiments were performed in whole zebrafish embryos with pharmacological calcium modulators like calcium chelators (BAPTA) (Webb et al., 1997; Chang and Lu, 2000) to validate these sensors in whole organisms, among others. I don’t understand why this kind of control experiments in whole zerbafish embryos expressing either camaleon has been shown.

My opinion is that the manuscript is acceptable for publication if deeper characterization of the sensor by control experiments is presented. These control experiments will increase the significance of results not only for this manuscript but of previous reports in which cameleon sensors have been used in zebrafish.

References.
Chang, D. C. and Lu, P. (2000) 'Multiple types of calcium signals are associated with cell division in zebrafish embryo', Microsc Res Tech 49(2): 111-22.
Ma, L. H., Webb, S. E., Chan, C. M., Zhang, J. and Miller, A. L. (2009) 'Establishment of a transitory dorsal-biased window of localized Ca2+ signaling in the superficial epithelium following the mid-blastula transition in zebrafish embryos', Dev Biol 327(1): 143-57.
Mizuno, H., Sassa, T., Higashijima, S., Okamoto, H. and Miyawaki, A. (2013) 'Transgenic zebrafish for ratiometric imaging of cytosolic and mitochondrial Ca2+ response in teleost embryo', Cell Calcium 54(3): 236-45.
Nagai, T., Ibata, K., Park, E. S., Kubota, M., Mikoshiba, K. and Miyawaki, A. (2002) 'A variant of yellow fluorescent protein with fast and efficient maturation for cell-biological applications', Nat Biotechnol 20(1): 87-90.
Nagai, T., Yamada, S., Tominaga, T., Ichikawa, M. and Miyawaki, A. (2004) 'Expanded dynamic range of fluorescent indicators for Ca(2+) by circularly permuted yellow fluorescent proteins', Proc Natl Acad Sci U S A 101(29): 10554-9.
Webb, S. E., Lee, K. W., Karplus, E. and Miller, A. L. (1997) 'Localized calcium transients accompany furrow positioning, propagation, and deepening during the early cleavage period of zebrafish embryos', Dev Biol 192(1): 78-92.

Comments for the author

No comments

Reviewer 2 ·

Basic reporting

The manuscript is clear and it was improved in comparison with previous versions. The authors have answer the reviewers comments. The five answers provided explain with clarity what changes were made to the work.

Experimental design

Experimental design is clear.

Validity of the findings

This work will be of interest for some researchers working in this field of research.

Comments for the author

The paper has been improved and in my opinion is ready for publication.

---

## Round 0.4 · Major Revisions

· Academic Editor

Major Revisions

Dear Sirs:

The point raised by the reviewer in his assessment of the paper is critical to address before the paper can be accepted for publication. Please address it.

·

Basic reporting

No comments

Experimental design

No comments

Validity of the findings

The use of different highly toxic compounds like bisphenol A or paraquat, are not adequate control treatments to characterize the specificity of the cameleon YC2.12 sensor. The observed sensor changes in response to these treatments can be through different mechanisms or signaling pathways like MAPK or G-proteins since these pathways are known to interact with ROS signaling and not only through calcium changes. The exposure time to paraquat, 25 hours, seems to be too long. This prolonged exposure could be inducing cell death and developmental alterations, as has been previously observed in other animal models like amphibians, when exposed to paraquat for much shorter times, like 2 hours (Mussi and Calcaterra 2010. Comparative Biochemistry and Physiology, Part C 151:240). In addition the paraquat concentration used by Tsuruwaka is very high 1mM, in contrast to the concentrations reported by other authors, including the references indicated in the manuscript by Tsuruwaka. For example in the P19 cell line the paraquat concentration is 1uM for shorter time, 6 hours (Shimada et al. 2016) and in zebrafish the paraquat treatment that induces autophagy range from 10uM to 1mM (Imamura et al. 2011). Even more in amphibians the paraquat concentration to induce evident developmental effects start at 10uM, a hundred times more diluted that the concentration used by Tsuruwaka. ¿Does that imply that cameleon YC2.12 signal changes are only observed under extreme treatments with high concentrations of toxic compounds like paraquat?
As I indicated since my first review, appropriate control experiments with known Ca2+ pharmacological modulators previously tested and reported in zebrafish should be used. Since that review I thought about this kind of compounds and not compounds that have not been fully characterized in zebrafish at these stages of development and that could be acting through other mechanisms and therefore in an indirect way. I think that the manuscript is acceptable only if the authors show that the cameleon sensor respond to known Ca2+ pharmacological modulators, not to extreme highly toxic compounds like paraquat.

Examples of pharmacological compounds.
BAPTA a calcium chelator (Chang, 2000. Microscopy research and technique 49: 111).
Thapsigargin (Schneider, et al. 2008. Development 135: 75-84).

---

## Round 0.5 · Minor Revisions

· Academic Editor

Minor Revisions

Please clarify the final issue brought by the reviewer.

·

Basic reporting

The manuscript was improved.

Experimental design

No comments

Validity of the findings

As shown by Popgeorgiev and collaborators in 2011, thapsigargin treatment of zebrafish embryos resulted in a signal increase of the calcium sensor Oregon Green BAPTA-1 compared to control embryos. In contrast, the results presented by Tsuruwaka and collaborators in Fig.S1, the thapsigargin treated embryos (B) show a decrease in the calcium sensor intensity compared to the control embryos (A). Tsuruwaka and collaborators should indicate how many embryos where analyzed in these experiments. In addition they should attempt to give a brief explanation of the potential reasons for the observed difference with the results previously reported by Popgeorgiev and collaborators as part of the discussion. I think that these new results add very relevant information about the response of the calcium sensor cameleon YC2.12 to thapsigargin treatment in whole living embryos, and make it of more interest to the research community working in calcium research and that are using different calcium sensors.

I consider that paying attention to these two details that I indicate above will complete the manuscript for publication.

---

## Round 0.6 · accepted · Accept

· Academic Editor

Accept

The revised manuscript is now acceptable for publication.